# Chronic RNA G-quadruplex accumulation in aging and Alzheimer's disease

**Lena Kallweit[1], Eric Daniel Hamlett[2], Hannah Saternos[3], Anah Gilmore[3], Ann-Charlotte Granholm[3]\*[†], Scott Horowitz[1]\*[†]**

[1]Department of Chemistry & Biochemistry and the Knoebel Institute for Healthy Aging, University of Denver, Denver, United States; [2]Department of Pathology and Laboratory Medicine, Medical University of South Carolina, Charleston, United States; [3]Department of Neurosurgery, University of Colorado Anschutz Medical Campus, Aurora, United States

---

## eLife Assessment

The current human tissue-based study provides **compelling** evidence correlating hippocampal expressions of RNA guanine-rich G-quadruplexes with aging and with Alzheimer's disease presence and severity. The results are **fundamental** and will rejuvenate our understanding of aging and AD's pathogenesis.

[Editors' note: this paper was reviewed by Review Commons.]

---

**\*For correspondence:**
ann-charlotte.granholm-bentley@cuanschutz.edu (A-CG);
scott.horowitz@du.edu (SH)

[†]Co-senior authors

**Competing interest:** The authors declare that no competing interests exist.

**Abstract** As the world population ages, new molecular targets in aging and Alzheimer's disease (AD) are needed to combat the expected influx of new AD cases. Until now, the role of RNA structure in aging and neurodegeneration has largely remained unexplored. In this study, we examined human hippocampal *postmortem* tissue for the formation of RNA G-quadruplexes (rG4s) in aging and AD. We found that rG4 immunostaining strongly increased in the hippocampus with both age and with AD severity. We further found that neurons with the accumulation of phospho-tau immunostaining contained rG4s, rG4 structure can drive tau aggregation, and rG4 staining density depended on APOE genotype in the human tissue examined. Combined with previous studies showing the dependence of rG4 structure on stress and the extreme power of rG4s at oligomerizing proteins, we propose a model of neurodegeneration in which chronic rG4 formation is linked to proteostasis collapse. These morphological findings suggest that further investigation of RNA structure in neurodegeneration is a critical avenue for future treatments and diagnoses.

## Introduction

As the aged population increases, Alzheimer's disease (AD) and AD-related dementias (ADRDs) will become one of the world's largest medical and economic crises due to the lack of early diagnostic methods and durable therapies (**GBD 2016 Neurology Collaborators, 2019**; **Breijyeh and Karaman, 2020**). Age is the highest risk factor for AD as disease prevalence increases exponentially as a function of age, and close to 40% of older adults over the age of 80 experience AD or ADRDs (**Armstrong, 2019**; **Hersi et al., 2017**). The most important genetic risk factor for developing late-onset AD (LOAD) is the apolipoprotein E (APOE) genotype, at least in Caucasians, where carriers of one or two copies of the APOE4 allele have a substantial increase in the risk for developing LOAD compared to carriers of the APOE2 or 3 alleles (**Hersi et al., 2017**; **Bejanin et al., 2021**). Beyond APOE, recent meta-analyses combining genome-wide association studies (GWAS) have expanded the number of AD-risk loci (**Novikova et al., 2021**), but most of the disease-associated variants reside in non-protein

coding regions of the genome, making it difficult to elucidate how they affect AD susceptibility and pathogenesis.

The exact mechanisms behind these two predominant AD risk factors remain a mystery but both are widely associated with increased abnormal protein aggregation, which is the most abundantly observed pathological hallmark of AD pathogenesis. AD and ADRDs all have an abnormal protein aggregation profile that includes (1) extracellular amyloid senile plaque deposition occurring at least 5–10 years before mild cognitive impairments (MCIs) are observed and (2) aggregates of phosphorylated Tau (p-Tau) forming intracellular neurofibrillary tangles (NFTs) (*Braak and Braak, 1995*; *Furman et al., 2017*). Novel biomarker studies have shown that hyperphosphorylated Tau species exist in plasma prior to the onset of dementia symptoms, with specific phosphorylation sites affecting the affinity of binding of the Tau protein to microtubules (*Gonzalez-Ortiz et al., 2023*). With advancing age, protein aggregation is accelerated by the accumulation of reactive oxygen species (ROS) as well as the onset of neuroinflammation in the brain (*Bennett et al., 2017*; *Ibáñez-Salazar et al., 2017*; *Leng and Edison, 2021*). Having APOE4 was shown to significantly increase accumulation of amyloid and NFTs in Caucasians and also increases inflammation and ROS in the human brain (*Bejanin et al., 2021*; *Raber et al., 2004*). The effects of these AD risk factors have been studied mostly at the protein level in the brain, but not yet at the RNA level, where structure and function have a myriad of important roles in numerous biological processes.

RNA higher-order structures of particular interest are RNA guanine-rich G-quadruplexes (rG4s) (*Dumas et al., 2021*). G-quadruplexes are secondary structures that can fold under physiological conditions by guanine-rich DNA and RNA molecules and occur in the genomes and transcriptomes of many species, including humans (*Rouleau et al., 2020*). rG4s are four-stranded helices formed by guanine tetrads through Hoogsteen base-pairing, with a monovalent cation channel running down the middle of the tetrads (*Figure 1a*; *Lyu et al., 2021*). G4s are potentially involved in the regulation of DNA and RNA processes, such as replication, transcription, translation, RNA localization, and degradation (*Bryan, 2019*; *Kejnovsky et al., 2015*; *Yuan et al., 2020*). Recently, rG4s were shown to form as a function of stress in eukaryotic cells (*Nakanishi and Seimiya, 2020*). Under nonstress conditions, rG4s were largely unfolded, but upon introduction of stress (including ROS stress) the rG4s preferentially folded and remained folded until the cessation of stress (*Kharel et al., 2023*). This observation raises the possibility that rG4s could preferentially fold under the increased and chronic stress conditions of aging and AD in the human brain, leading to abnormally high levels of rG4 formation in older adults or in adults with protein misfolding diseases. RNA has been known to be included in AD-related aggregates of both amyloid and NFTs in patients with AD for decades (*Wolozin and Apicco, 2015*), and a recent sequencing study of nucleic acids embedded in AD aggregates showed an enrichment in potential rG4-forming sequences (*Shmookler Reis et al., 2021*).

It has recently been shown that rG4s can greatly enhance protein oligomerization (*Masai et al., 2019*; *Begeman et al., 2020*). Moreover, rG4s can bind to and greatly enhance the phase transition of tau (*Yabuki et al., 2024*). Similarly, Zwierzchowski-Zarate et al. explored the effects of different RNA sequences on tau aggregation in a cellular model (*Zwierzchowski-Zarate et al., 2022*). In their study, several sequences caused an increase in tau aggregation, but the highest level of tau aggregation was achieved with a GGGC repeat sequence (*Zwierzchowski-Zarate et al., 2022*). However, the structure of the RNAs, including possible rG4 formation, was not explored or speculated upon. We therefore performed CD spectroscopy to determine if the sequence associated with the highest levels of tau fibrillation might be forming rG4 secondary structures. The CD spectra show that this RNA forms parallel rG4 structures in vitro (*Figure 1—figure supplement 1*).

Based on these previous in vitro and cellular studies, an increased presence of rG4s in the brain could contribute to AD-associated aggregation. However, the presence of rG4s in the human brain or other human tissues has not been shown previously, nor how it relates to aging or AD pathology. This was the focus of the current study.

## Results
### Formation of rG4s in aging

To determine if rG4s form in the human brain, we used the well-characterized single-chain BG4 antiquadruplex antibody (Sigma-Aldrich MABE917, antibody null controls in *Figure 1—figure supplement*

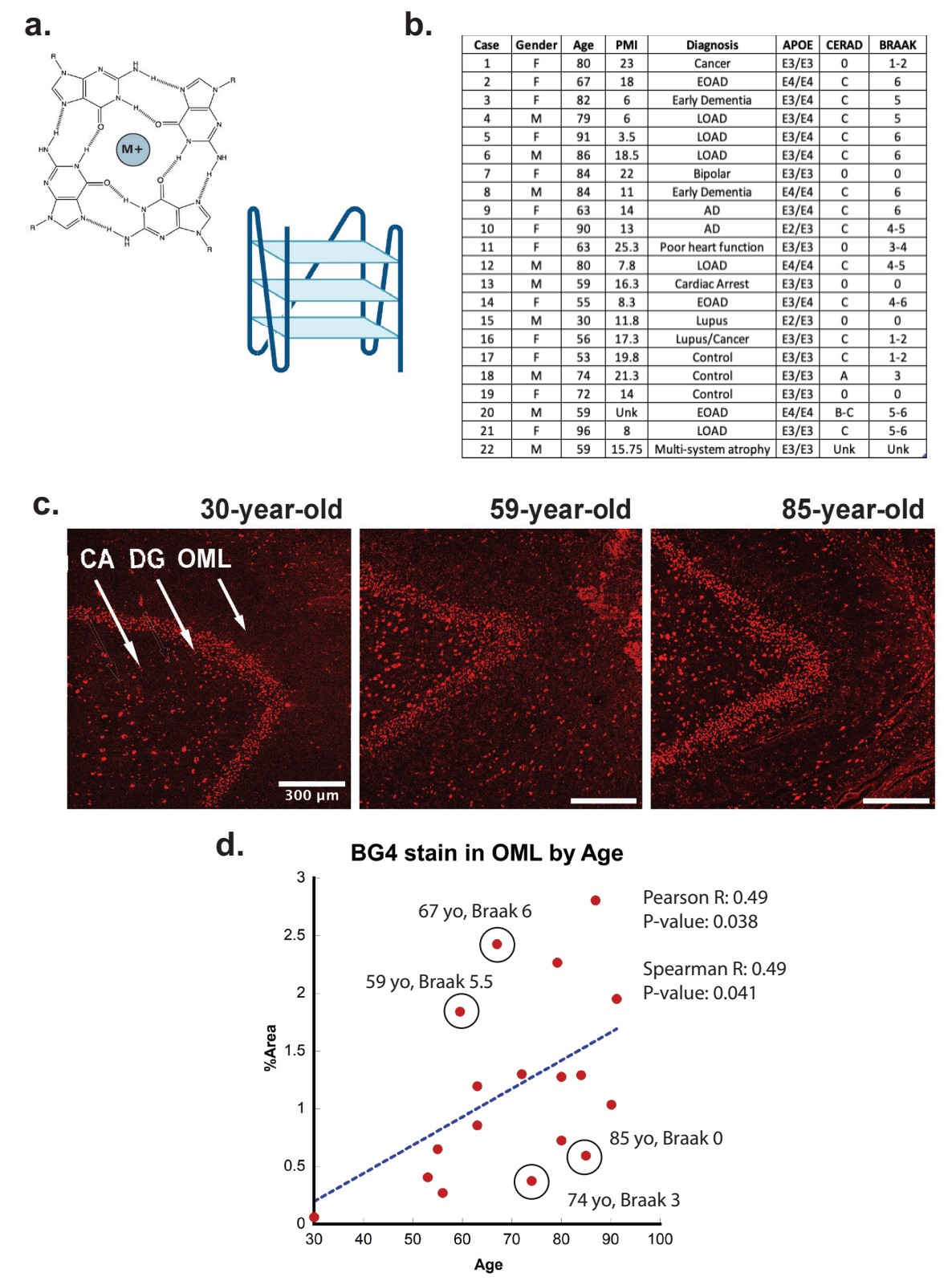

**Figure 1.** G-quadruplex correlation with age. (**a**) (Left) Representative G-quadruplex tetrad structure with Hoogsteen base pairing stabilized by a cation at the center. (Right) G-quadruplex structure with stacked tetrads. (**b**) Demographics table for individuals included in this study. (**c**) Immunofluorescence showing BG4 staining (red) in brain sections of human hippocampus from four control individuals of increasing age ranging 30–91 years old (cases 15,

*Figure 1 continued on next page*

*Figure 1 continued*

13, and 8, respectively) all with Braak stage 0. (**d**) Correlation of percent area covered by BG4 fluorescence in the outer-molecular layer versus age of individual.

The online version of this article includes the following figure supplement(s) for figure 1:

**Figure supplement 1.** Circular dichroism spectra showing G-quadruplex structure of RNA sequence able to most efficiently template tau fibrillation in *Zwierzchowski-Zarate et al., 2022*.

**Figure supplement 2.** Antibody null controls.

**Figure supplement 3.** No significant correlation observed between BG4 stain and postmortem interval (PMI).

**Figure supplement 4.** ImageJ quantification.

*2*) to stain human hippocampal *postmortem* tissue of different ages under conditions that strongly favor cytoplasmic rG4 identification using 4% paraformaldehyde fixation in cells (*Laguerre et al., 2016*; *Mestre-Fos et al., 2020*). The tissues were obtained in collaboration with the brain bank at the Medical University of South Carolina. The study included 22 cases, male and female, of different ages and with or without a clinical or neuropathological diagnosis of AD (see *Figure 1b*). All cases had received a neuropathological examination including assessment of Braak and Consortium to Establish a Registry for Alzheimer's Disease (CERAD) stages according to the updated NIA criteria except in those cases in which no pathology was noted (*Braak and Braak, 1995*; *Jack et al., 2012*).

First, the rG4 staining profiles were compared between different ages, with an age range of 30–92 years of age (*Figure 1c*). We discovered a significant age-related increase in rG4 immunostaining, with the densest staining observed in older individuals (*Figure 1c and d*). The largest difference in staining between young and older cases was observed in the outer molecular layer (OML) and in the hilar region, CA4 (see *Figure 1c*, legend arrows). The strongest overall staining for BG4 was found in the dentate gyrus granular cell layer (DG), but with less observable age-related increases. Staining density measurements confirmed these findings and demonstrated a highly significant increase in BG4 staining in the oldest cases studied within the OML (*Figure 1d*) and the CA4 regions of cornu ammonis (*Figure 2—figure supplement 2a*), confirming increased intracellular BG4 accumulation as a function of age. Plotting BG4 percent area versus age confirmed the observed findings and showed correlation (Spearman $R = 0.49$) with high significance (p=0.038 in the OML region (*Figure 1d*). Positive and significant Pearson but not Spearman correlations were observed between age and BG4 area in the CA4 region, *Figure 2—figure supplement 2a*), and nonsignificant correlations were observed in the dentate granule cell layer. No significant correlation was found with BG4 stain and *postmortem* interval (PMI) (Spearman $R = –0.32$, p=0.2 see *Figure 1—figure supplement 3*). In total, these data show that rG4s increased as a function of age in the human brain. We observed multiple outliers in both older and younger individuals due to Braak stage. For example, an 85-year-old individual with Braak stage 0 or a 59-year-old individual with Braak stage 5.5 were outliers as a function of age. This led us to analyze whether rG4 formation increased with Braak stage, a measure of AD severity.

## Formation of rG4s in AD

Next, we examined whether the formation of rG4s was associated with AD neuropathology. Braak staging is a neuropathological staging method that takes into account the density of NFTs both within each brain region and also from region to region (*Trejo-Lopez et al., 2022*), with higher numerical values representing greater AD severity (*Braak and Braak, 1995*). Like in the case of aging, increased BG4 staining was readily apparent in AD cases, especially in those cases associated with a higher Braak stage (*Figure 2a*). In the OML, plotting BG4 percent area versus Braak stage demonstrated a strong correlation (Spearman $R = 0.72$) with highly significantly increased BG4 staining with higher Braak stages (p=0.00086) (*Figure 2b*). Positive significant correlations ($R = 0.52$, p=0.028) between Braak stage severity and BG4 staining were also observed within the CA4 region (*Figure 2—figure supplement 2*). Interestingly, the OML region is highly involved in AD pathology and is a hippocampal region where AD pathology is often observed, (*Dong et al., 2007*), suggesting that the presence of rG4s in this particular brain region might have significance for amyloid as well as NFT formation during the AD disease process. In addition, a prominent loss of synapses in AD is found in the OML, suggesting a potential close connection with layer 2 of the entorhinal cortex (*Lassmann et al., 1993*).

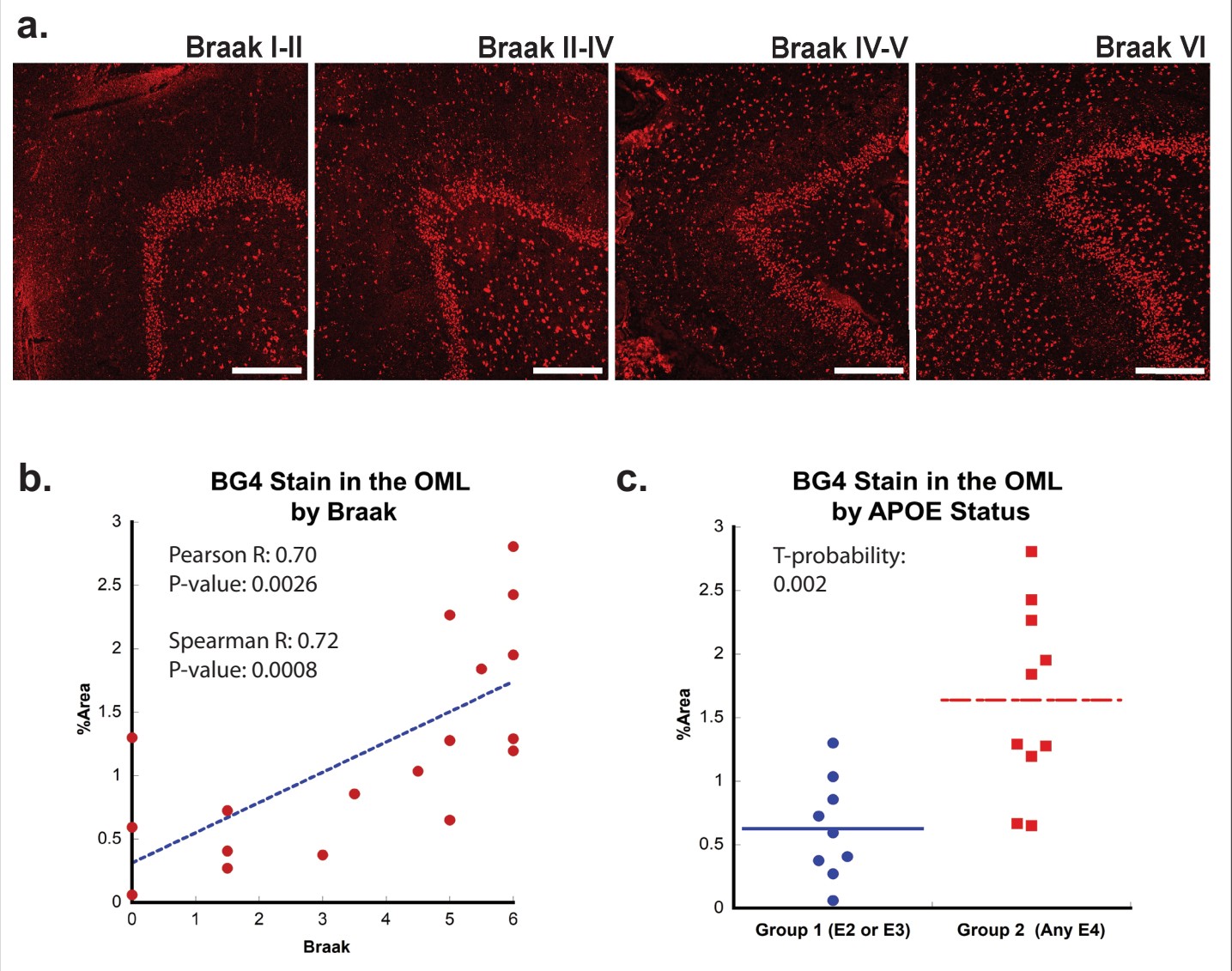

**Figure 2.** Effect of Braak stage or ApoE status on G-quadruplex level. (**a**) Immunofluorescence showing BG4 staining (red) in brain sections of human hippocampus with increasing Braak stage (1–6) associated with Alzheimer's disease (AD) severity cCases 16, 11, 10, and 8, respectively. All scale bars are 300 μm. (**b**) Correlation of percent area BG4 coverage in the outer molecular layer versus severity of Braak stage. (**c**) Quantification of percent area BG4 coverage by population with E2/E3 and E3/E3 versus E3/E4 and E4/E4 apolipoprotein E (APOE) alleles.

The online version of this article includes the following figure supplement(s) for figure 2:

**Figure supplement 1.** Very weak correlation was observed between age and Braak stage.

**Figure supplement 2.** Regression of Age and Braak stage.

If rG4 formation is more prevalent in AD than in non-AD cases and directly involved in AD pathology, we would also predict that rG4 formation would be more prevalent in APOE genotypes that are predisposed for higher LOAD risk, that is, patients with one or two copies of the APOE4 genotype. Indeed, there was a significant (*t*-probability 0.002) increase in BG4 staining levels in the OML in *post-mortem* cases with one or more APOE4 allele compared to those with only APOE2 or APOE3 alleles (*Figure 2c*). This result, although correlative, is consistent with a previous study performed in vitro in which the RNA sequences found in aggregates in different APOE genotype cell lines were different (*Shmookler Reis et al., 2021*).

## Patterns of rG4 localization

Confocal microscopy images showed that a large amount of the BG4 immunostaining was localized in the cytoplasm near the nucleus in a punctate pattern, reminiscent of known rG4 formation in vitro and in cell lines in rRNA ribosomal extensions (*Mestre-Fos et al., 2020*). We therefore also stained with an anti-rRNA antibody to determine whether this was the case in human tissue as well. As expected, the stains overlapped (*Figure 3b*, *Figure 3—figure supplement 1*), showing that at least a large percentage of the observed BG4 signal arose from rG4s and not nuclear DNA G4s. This result also confirms that previously observed rG4 formation in ribosomes (*Mestre-Fos et al., 2020*) also occurs in human brain tissue.

To further explore the formation of rG4s in the human hippocampus, we examined the staining pattern in different cell types based on specific antibody markers. Co-staining with markers for the most common brain cell types, rG4s were clearly abundant in neurons, oligodendrocytes, and astrocytes (*Figure 3c*). rG4s exhibited no apparent colocalization with microglia (*Figure 3c*). This pattern is especially noticeable in the double labeling with SOX10 (*Figure 3c*), which shows the nuclear oligodendrocyte marker SOX10 (*Turnescu et al., 2018*) staining in proximity to the cytoplasmic BG4 immunostaining.

We co-stained BG4 with different p-Tau epitope antibodies to determine whether pathological aggregation of Tau proteins co-occurred with BG4 immunostaining (*Figure 4*). Indeed, we found that BG4 immunostaining co-localized with p-Tau immunostaining in neurons located in the CA1–4 regions of the hippocampus (*Figure 4*). We used two different antibodies directed against the serine 396 (S396) and the threonine 231 (T231) phosphorylation sites to investigate co-localization with different p-Tau epitopes. Quantifying the percentage of cells co-stained with BG4 and p-Tau revealed that 85% of the S396 p-Tau-positive cells also had positive BG4 staining, and 82% of the T231 p-Tau-positive cells also had positive BG4 staining.

## Discussion

In this study, we demonstrated that rG4s form preferentially as a function of age and AD pathology in neurons and glial cells in the human *postmortem* brain. While the granule cell layer in the dentate gyrus exhibited the most BG4 immunostaining overall, this was not the area most changed with aging and AD as considerably greater changes were observed in the OML and the CA4 region of the hippocampus. rG4s were primarily found in the cytoplasm near the nucleus of neurons, astrocytes, and oligodendrocytes and were also found in neurons that contained regions of positive staining against two different epitopes of p-Tau. The BG4 staining levels were also impacted by APOE status, with the presence of one or two APOE4 alleles associated with higher BG4 levels in the OML region. The OML is an interesting region of the hippocampus from the pathology standpoint since this is the region of the hippocampus exhibiting the most prominent synaptic loss early in the disease process (*Lassmann et al., 1993*; *Haytural et al., 2021*). We have recently demonstrated a significant loss of presynaptic components of the OML in AD (*Haytural et al., 2021*), strengthening the findings observed herein and suggesting a particular vulnerability of this region of the hippocampus to AD progressive pathology.

Combining these observations with those previously made on the reliance of rG4 structure on cellular stress and the roles of rG4s in protein oligomerization and tau aggregation in vitro, we can propose a model of how rG4s form and how they could perpetuate the effects of AD (*Figure 4c*). As ROS and other protein stresses accumulate with age and according to APOE genotype, rG4s form. Although rG4s can rescue protein folding and could release proteins when unfolded at stress cessation (*Son et al., 2023*), under persistent stress RNA would not be released from the G-quadruplexes, leading to increased protein oligomerization and aggregation (*Begeman et al., 2020*). This function has previously been suggested for RNA in biomolecular condensates more generally (*Mateju et al., 2017*). The acceleration of protein oligomerization and aggregation by the rG4s could then contribute to a vicious cycle of increased aggregation, leading to synaptic loss as described recently for the OML (*Haytural et al., 2021*). Recent in vitro work confirms an interaction between rG4 formation and p-Tau aggregation specifically (*Yabuki et al., 2024*). In vitro experiments focused on the specific mechanisms for the rG4 influence on Tau phosphorylation and/or aggregation will be an important future focus following these preliminary observations to more clearly delineate the roles of rG4s in neurodegeneration. Of note, the previous studies showing that the fixation conditions used here

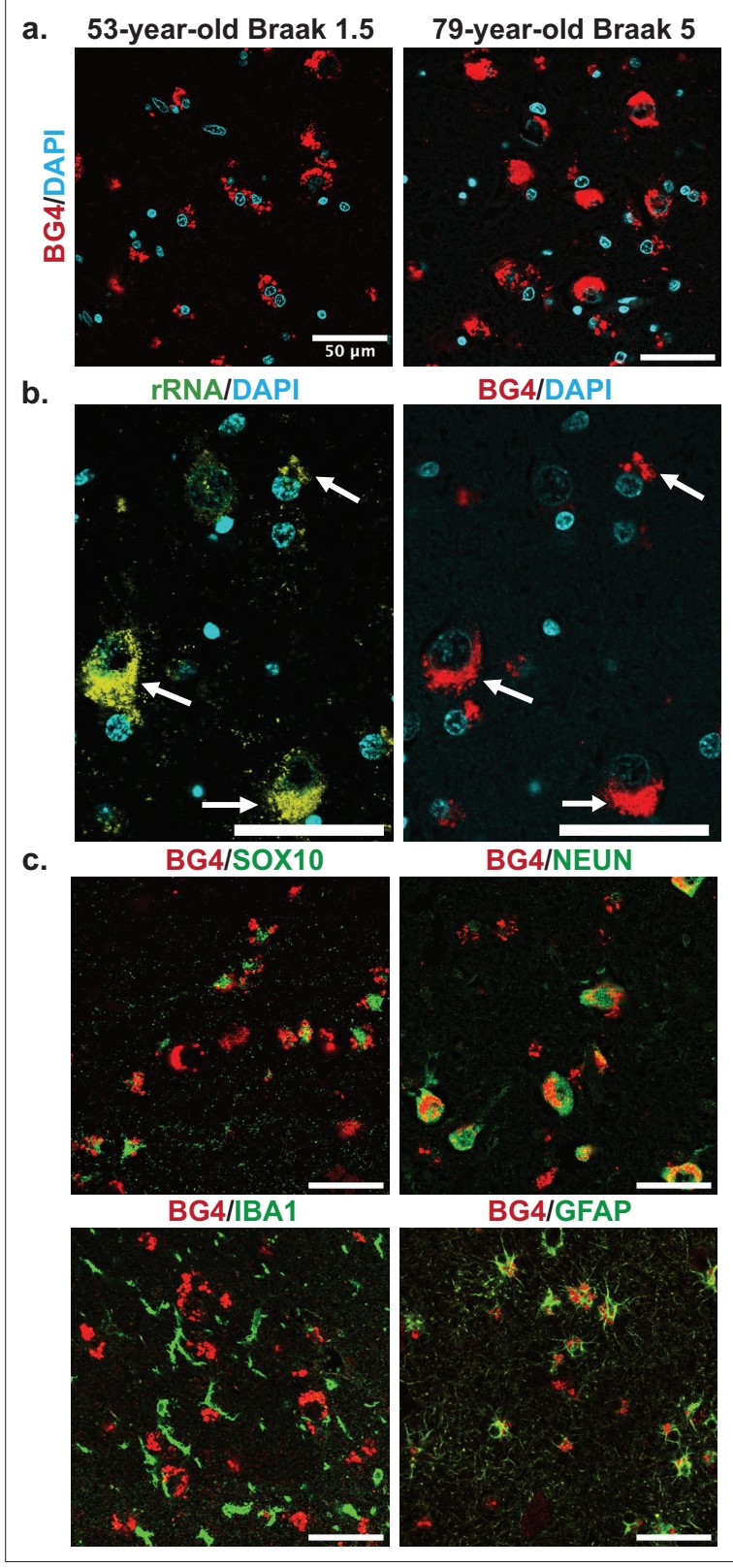

**Figure 3.** Cellular types and localization of rG4s, all scale bars are 50 µm. (**a**) Immunofluorescence showing BG4 staining patterns (red) in human hippocampal tissue of case 17, a 53-year-old individual with Braak stage 1.5, and case 4, a 79-year-old individual with Braak stage 5 Alzheimer's disease (AD) pathology (right). Cell nuclei stained with DAPI (blue). (**b**) Co-stain of rRNA (left, yellow green) and BG4 (right, red), in the hippocampal CA4

*Figure 3 continued*

region, an 80-year-old individual with early-stage tauopathy, demonstrating significant but not complete overlap between BG4 and rRNA. (**c**) BG4 staining (red) with cell type markers (green) of oligodendrocytes (SOX10, case 8), neurons (NEUN, case 4), microglia (IBA1, case 18), and astrocytes (GFAP, case 18). BG4 is most prevalent in oligodendrocytes, neurons, and astrocytes. BG4 does not strongly colocalize with microglia (IBA1).

The online version of this article includes the following figure supplement(s) for figure 3:

**Figure supplement 1.** Ribosomal RNA antibody.

strongly favor rG4 identification instead of DNA G4 identification were performed in cells, and so it is possible that in tissue some DNA staining could also occur; however, this is not suggested by the co-stain with rRNA (*Figure 3b*).

Of note, the accumulation or rG4s in neurodegeneration could have roles in addition to those of protein aggregation. RNA transport dysregulation has also been a common factor identified in neurodegeneration (*Fernandopulle et al., 2021*), and it could be possible that the rG4 build-up observed here could be a product of this process or could lead to it. It has also been proposed that the RNA transport dysregulation and protein aggregation processes could be linked (*Bitetto and Di Fonzo, 2020*). This remains an area for further investigation.

This work raises the possibility of using rG4s as potential drug targets in AD, or given the high correlation with AD severity, as early AD biomarkers. Future work can focus on ameliorating AD symptoms through rG4 binding molecules or detecting brain-derived rG4s as disease biomarkers for diagnosis. Many different rG4-binding molecules have already been created and shown to bind in human cells (*Zheng et al., 2023*) and provide a set of promising lead compounds for potential AD treatment.

Although here we have examined only AD, there are many other aging-related diseases featuring protein aggregation, including Parkinson's disease, ALS, Fragile X syndrome, and Huntington's disease (*Ross and Poirier, 2004*). Notably, all of these diseases now have significant evidence showing rG4s as interactors with the aggregating proteins (*Imperatore et al., 2020*; *Ishiguro et al., 2020*; *Liu and Xu, 2018*; *Mori et al., 2013*; *Oyoshi and Kurokawa, 2012*; *Scalabrin et al., 2017*; *Takahama et al., 2011*; *Liu et al., 2021*; *Mejzini et al., 2019*; *Guerrero et al., 2016*; *Riccardi et al., 2022*; *Goering et al., 2020*; *Matsuo et al., 2023*), and in the case of Fragile X, a recent study demonstrated that rG4s could contribute to its aggregation and neurotoxicity (*Asamitsu et al., 2021*). In addition, a recent manuscript by Raguseo et al. showed an increased prevalence of G4 structures in C9orf72 mutant human motor neurons in ALS/FTD compared to healthy motor neurons, and direct involvement of the rG4 complex in the aggregation process (*Raguseo et al., 2023*). These recent findings illustrate the need to explore rG4 complexes and their role in neurodegeneration further. The results here combined with these previous studies suggest that the formation of rG4s could be a common mechanism of neurodegeneration.

## Materials and methods

### Human brain tissue

This work was approved by both the University of Colorado and Medical University of South Carolina Institutional Review Boards (IRB), and the study was performed in accordance with the ethical standards of the 1964 Declaration of Helsinki and its later amendments. The University of Colorado IRB protocol was approved in January 2022 (#21-4648) and was reviewed by the UCD Panel D COMIRB review committee. The Medical University of South Carolina IRB protocol was approved in August 2019 (#00079529) and was reviewed by the IRB I review committee. Based on these reviews, this project met the Not Human Research criteria set forth by the Code of Federal Regulations (45CFR46). Since the use of postmortem brain tissue for research is exempt according to federal IRB regulations and does not require a full IRB review and approval, each laboratory that participated in the research of human brain tissue had a valid non-human research (NHR) protocol.

The next of kin, in the absence of actual contrary indications by the decedent, gave the decedent's brain and associated structures for educational and research purposes. The next of kin signed a postmortem consent form and the original of this postmortem consent form is kept in a locked drawer and the identities are never revealed to researchers studying the brain tissue and associated biofluids. All efforts were made to consider diversity, equity, and inclusion in the selection of brain tissues for this

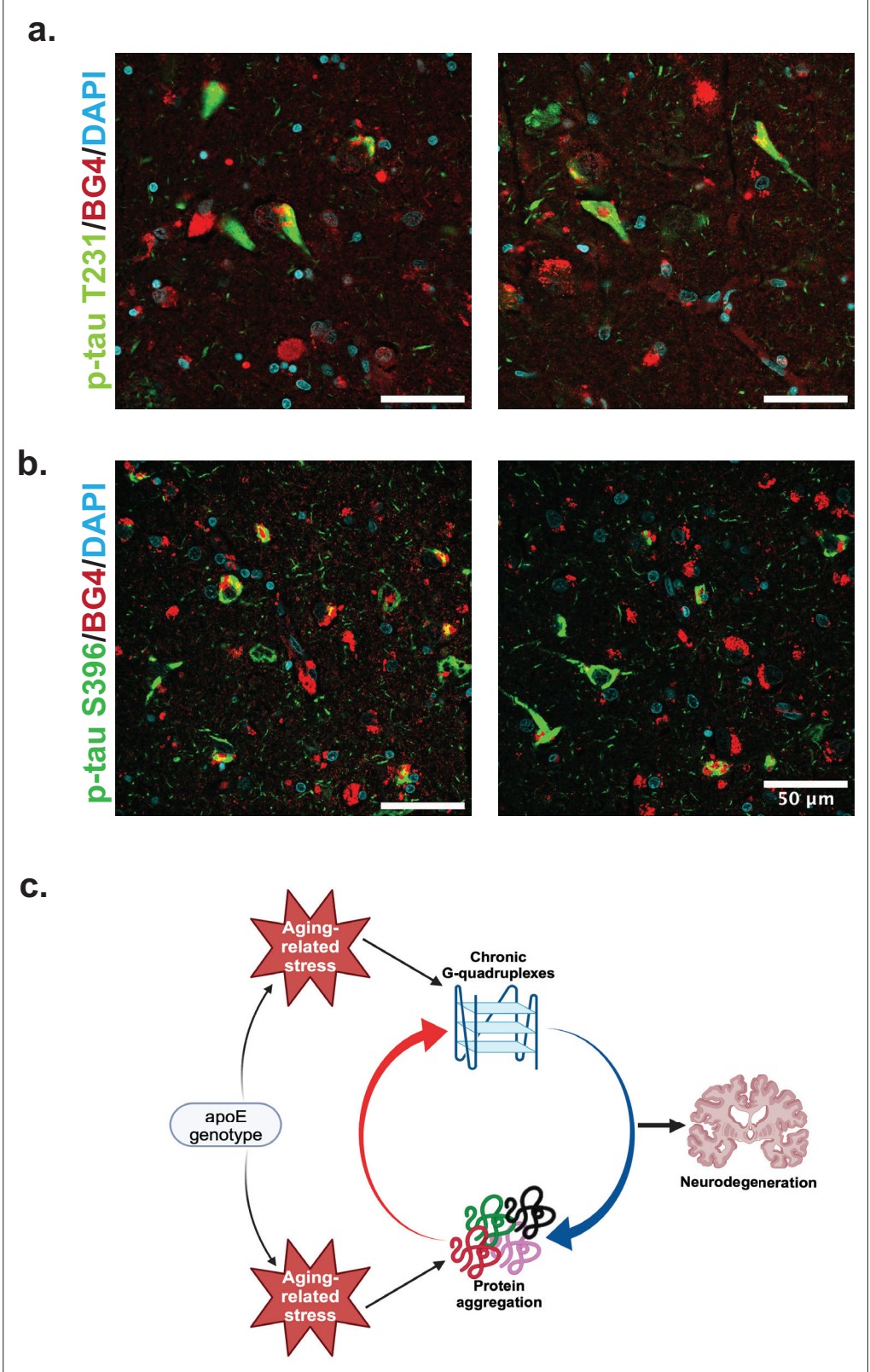

**Figure 4.** Colocalization of rG4s with p-tau and neurodegeneration model. (**a**) Immunofluorescence of human hippocampus from two older Alzheimer's disease (AD) individuals (cases 8 and 21) showing BG4 (red), DAPI (blue), and p-Tau (green). (**b**) Proposed model relating chronic rG4 formation and protein aggregation in aging and neurodegeneration.

study. *Figure 1b* shows the demographics for the 21 cases used for this study. The tissue was sliced in 1 cm coronal slices and subjected to free-floating fixation in 4% paraformaldehyde in phosphate-buffered saline (PBS) for 72 hours. The fixed tissue was then transferred to a 30% sucrose in phosphate buffer (PB), followed by transfer to a cryoprotection solution in PBS (30% glycerol, 30% ethylene glycol, and 40% $PO_4$ buffer) and stored at –20°C until dissection. Anatomical blocks from 19 brain regions were dissected from the 1 cm slabs, embedded in paraffin, and cut at a thickness of 5 µm on a Microm Microtome (Thermo Fisher Scientific Inc, Waltham, MA) and stained for routine neuropathological staging according to the NIA-AA *Revised Criteria for Diagnosis and Staging of Alzheimer's Disease* (*Hyman et al., 2012*).

## Immunofluorescence

Paraffin-embedded sections were deparaffinized in xylene (RPI, Cat#111056) and rehydrated using wash steps at 100%, 95%, 70%, and 50% ethanol followed by distilled water. Heat-induced antigen retrieval was performed using 0.05% citraconic acid at pH 7.2 for 20 minutes in a steamer. Slides were then washed with a saline solution buffered with tris(hydroxymethyl) aminomethane (Tris, TBS) and blocked with 0.25% triton-X and 5% normal serum for 1 hour. After blocking, slides were rinsed again with TBS and incubated with 1X True Black Lipofuscin Autofluorescence Quencher (Biotium Cat#23007) diluted in 70% ethanol. Slides were washed three times with TBS then incubated with primary antibodies listed in *Supplementary file 1* overnight, followed by goat anti-mouse AlexaFluor 555 (Thermo Scientific #A32727) or Donkey anti-rabbit Alexafluor 488 (Invitrogen #A-21206) at 1:500 dilution for 1 hour. Slides were then washed three times with TBS and cover-slipped using mounting media containing DAPI (Thermo Scientific, Cat#P36931). Staining controls included omission of primary or secondary antibodies (*Figure 1—figure supplement 2*). In *Figure 3b*, we performed sequential double labeling, first with BG4, and then with rRNA antibody at 1:250 (Abcam #ab17119).

In addition, we examined whether Braak stage correlated with aging itself (*Figure 2—figure supplement 1*). Although not significant at the 0.05 level, there was a weak correlation between Braak and Age. Braak staging is a paradigm used to describe density and regional spread of NFTs in the Alzheimer brain and includes levels 0-VI. CERAD is a staging paradigm for AD that takes into account amyloid plaque formation (*Trejo-Lopez et al., 2022*).

## Imaging and statistical methods

Slides were imaged at high magnification using an Olympus confocal laser scanning microscope equipped with FlowView FV3000 software (Olympus, Tokyo, Japan) with 16-bit grayscale with 65,000 shades of grey at ×10, ×20, and ×60 magnification at wavelengths 355, 488, and 555 m. Images used for quantification were taken at constant microscope and intensity settings. All image quantification was performed using Fiji ImageJ. Quantification was performed by a researcher who was blinded to the neuropathological diagnosis of individual cases and with no knowledge of Braak stage or age for the case being quantified. *Supplementary file 2* presents side-by-side comparisons of quantification from two researchers who were both blinded to group diagnosis and individual cases. The data used in the paper are highlighted. One data point with too much variation between quantifications was removed to increase rigor. Background was subtracted using a rolling ball radius. Images were then divided into three regions of interests (ROIs); OML, DG, and CA4 (*Figure 1—figure supplement 3*). Images were submitted to an autothreshold and particles analyzed to get a percent area measurement of BG4 staining. These were graphed and linearly fit using Kaleidagraph using Pearson fit. Spearman and Pearson *R* and p-values are reported for 18 cases for age correlation and 18 cases for Braak correlation. An unpaired *t*-test with unequal variance was performed for the two groups E2/E3 and E3/E3 versus E3/E4 and E4/E4. When a single outlier based on the singular African American case was removed from the APOE quantification, the *t*-probability remained highly significant at 0.0045 versus 0.002. All quantification was performed on ×10 images obtained from an Olympus microscope.

## Circular dichroism

CD spectra were obtained using a Jasco J-1100 circular dichroism at 25°C. Sequence was resuspended in 10 mM potassium phosphate pH 7.5 buffer and diluted to 12.5 μM (per strand) RNA. The CD measurement was taken from 300 nm to 210 nm at 1 nm intervals using a 50 nm/min scanning speed. The sequence used was 5'-CGGGCGGCGGGGGGGGCCCGGGCGGCGGGGGGGGCCCGGGCG-3' (*Zwierzchowski-Zarate et al., 2022*). Parallel structure was determined using previously described quantification methods for quadruplex topology (*Cheng et al., 2018*).

## Acknowledgements

Funding for this project was provided by NIH R35GM142442 to SH and NIH 5R01AG071228-02, R01AG070153, and R01AG061566, as well as a grant from the BrightFocus Foundation (CA2018010) and a grant from Lejeune Foundation

(GRT-2023b/2277) to ACG.

## Additional information

### Funding

| Funder | Grant reference number | Author |
| --- | --- | --- |
| National Institute of General Medical Sciences | R35GM142442 | Scott Horowitz |
| National Institute on Aging | 5R01AG071228-02 | Ann-Charlotte Granholm |
| National Institute on Aging | R01AG070153 | Ann-Charlotte Granholm |
| National Institute on Aging | R01AG061566 | Ann-Charlotte Granholm |
| BrightFocus Foundation | CA2018010 | Ann-Charlotte Granholm |
| Lejeune Foundation | GRT-2023b/2277 | Ann-Charlotte Granholm |

The funders had no role in study design, data collection and interpretation, or the decision to submit the work for publication.

### Author contributions

Lena Kallweit, Data curation, Validation, Investigation, Methodology, Writing – original draft, Writing – review and editing; Eric Daniel Hamlett, Data curation, Investigation, Methodology, Writing – review and editing; Hannah Saternos, Investigation, Methodology, Writing – review and editing; Anah Gilmore, Data curation, Methodology, Writing – review and editing; Ann-Charlotte Granholm, Resources, Data curation, Supervision, Funding acquisition, Methodology, Writing – original draft, Project administration, Writing – review and editing; Scott Horowitz, Conceptualization, Supervision, Funding acquisition, Visualization, Methodology, Writing – original draft, Project administration, Writing – review and editing

### Author ORCIDs

Lena Kallweit ⓘ https://orcid.org/0009-0007-2452-8300
Eric Daniel Hamlett ⓘ https://orcid.org/0000-0002-2900-7949
Ann-Charlotte Granholm ⓘ https://orcid.org/0000-0002-9685-7599
Scott Horowitz ⓘ https://orcid.org/0000-0002-1148-0105

### Ethics

This research was conducted with strict adherence to ethical principles, ensuring all human subjects or legal next-of-kin provided informed consent, had the right to withdraw at any time, and their privacy and confidentiality were protected throughout the study, in accordance with institutional and national guidelines for human subject research.

Reviewer #1 (Public review): https://doi.org/10.7554/eLife.105446.3.sa1

Reviewer #2 (Public review): https://doi.org/10.7554/eLife.105446.3.sa2
Author response https://doi.org/10.7554/eLife.105446.3.sa3

## Additional files

### Supplementary files

Supplementary file 1. Antibodies used in the study and their respective dilutions and sources.

Supplementary file 2. Quantification of BG4 stain by % area in the OML performed by three different individuals, two of which were completely blind to the case information. Highlighted column is the quantification included in the main text. All three *R* values are highly significant.

MDAR checklist

### Data availability

All raw images have been deposited to the Open Science Framework under accession code BV8FP.

The following dataset was generated:

| Author(s) | Year | Dataset title | Dataset URL | Database and Identifier |
|---|---|---|---|---|
| Kallweit L | 2025 | 2025 RNA G4 eLife paper | https://doi.org/10.17605/OSF.IO/BV8FP | Open Science Framework, 10.17605/OSF.IO/BV8FP |

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
