## [Editor Report · eLife Assessment]

The current human tissue-based study provides **compelling** evidence correlating hippocampal expressions of RNA guanine-rich G-quadruplexes with aging and with Alzheimer's disease presence and severity. The results are **fundamental** and will rejuvenate our understanding of aging and AD's pathogenesis.

[Editors' note: this paper was reviewed by Review Commons.]

---

## [Referee Report · Reviewer #1 (Public review)]

This is an interesting manuscript where the authors systematically measure rG4 levels in brain samples at different ages of patients affected by AD. To the best of my knowledge this is the first time that BG4 staining is used in this context and the authors provide compelling evidence to show an association with BG4 staining and age or AD progression, which interestingly indicates that such RNA structure might play a role in regulating protein homeostasis as previously speculated. The methods used and the results reported seems robust and reproducible.

---

## [Referee Report · Reviewer #2 (Public review)]

RNA guanine-rich G-quadruplexes (rG4s) are non-canonical higher order nucleic acid structures that can form under physiological conditions. Interestingly, cellular stress is positively correlated with rG4 induction.

In this study, the authors examined human hippocampal postmortem tissue for the formation ofrG4s in aging and Alzheimer Disease (AD). rG4 immunostaining strongly increased in the hippocampus with both age and with AD severity. 21 cases were used in this study (age range 30-92).

This immunostaining co-localized with hyper-phosphorylated tau immunostaining in neurons. The BG4 staining levels were also impacted by APOE status. rG4 structure was previously found to drive tau aggregation. Based on these observations, the authors propose a model of neurodegeneration in which chronic rG4 formation drives proteostasis collapse.

This model is interesting, and would explain different observations (e.g., RNA is present in AD aggregates and rG4s can enhance protein oligomerization and tau aggregation).

---

## [Author Response]

The following is the authors’ response to the original reviews.

**Public Reviews:**

**Reviewer #1 (Public review):**
This is an interesting manuscript where the authors systematically measure rG4 levels in brain samples at different ages of patients affected by AD. To the best of my knowledge this is the first time that BG4 staining is used in this context and the authors provide compelling evidence to show an association with BG4 staining and age or AD progression, which interestingly indicates that such RNA structure might play a role in regulating protein homeostasis as previously speculated. The methods used and the results reported seems robust and reproducible. There were two main things that needed addressing:(1) Usually in BG4 staining experiments to ensure that the signal detected is genuinely due to rG4 an RNase treatment experiment is performed. This does not have to be extended to all the samples presented but having a couple of controls where the authors observe loss of staining upon RNase treatment will be key to ensure with confidence that rG4s are detected under the experimental conditions. This is particularly relevant for this brain tissue samples where BG4 staining has never been performed before.(2) The authors have an association between rG4-formation and age/disease progression. They also observe distribution dependency of this, which is great. However, this is still an association which does not allow the model to be supported. This is not something that can be fixed with an easy experiment and it is what it is, but my point is that the narrative of the manuscript should be more fair and reflect the fact that, although interesting, what the authors are observing is a simple correlation. They should still go ahead and propose a model for it, but they should be more balanced in the conclusion and do not imply that this evidence is sufficient to demonstrate the proposed model. It is absolutely fine to refer to the literature and comment on the fact that similar observations have been reported and this is in line with those, but still this is not an ultimate demonstration.Comments on current version:The authors have now addressed my concerns.

We thank the reviewer for their support!

**Reviewer #2 (Public review):**
RNA guanine-rich G-quadruplexes (rG4s) are non-canonical higher order nucleic acid structures that can form under physiological conditions. Interestingly, cellular stress is positively correlated with rG4 induction.In this study, the authors examined human hippocampal postmortem tissue for the formation ofrG4s in aging and Alzheimer Disease (AD). rG4 immunostaining strongly increased in the hippocampus with both age and with AD severity. 21 cases were used in this study (age range 30-92).This immunostaining co-localized with hyper-phosphorylated tau immunostaining in neurons. The BG4 staining levels were also impacted by APOE status. rG4 structure was previously found to drive tau aggregation. Based on these observations, the authors propose a model of neurodegeneration in which chronic rG4 formation drives proteostasis collapse.This model is interesting, and would explain different observations (e.g., RNA is present in AD aggregates and rG4s can enhance protein oligomerization and tau aggregation).Main issue from the previous round of review:There is indeed a positive correlation between Braak stage severity and BG4 staining, but this correlation is relatively weak and borderline significant (R = 0.52, p value = 0.028). This is probably the main limitation of this study, which should be clearly acknowledged (together with a reminder that "correlation is not causality"). Related to this, here is no clear justification to exclude the four individuals in Fig 1d (without them R increases to 0.78). Please remove this statement. On the other hand, the difference based on APOE status is more striking.Comments on current version:The authors have made laudable efforts to address the criticisms I made in my evaluation of the original manuscript.

We thank the reviewer for their support!

**Recommendations for the authors:**

**Reviewing Editor:**
I would suggest two minor edits:- The findings are correlative and descriptive, but the title implies functionality (A New Role for RNA G-quadruplexes in Aging and Alzheimer′s Disease). I would suggest toning down this title.- While I understand the limitations in performing additional biochemical experiments to validate the immunofluorescence study, I think this is worth mentioning as a limitation in the text.

We have made these two changes as requested, altering the title to remove the word Role that may imply more meaning than intended, and adding a line to the discussion on the need for future additional biochemical experiments.

**Reviewer #1 (Recommendations for the authors):**
Thanks for addressing the concerns raised.

We thank the reviewer for their support!

**Reviewer #2 (Recommendations for the authors):**
Minor point:Related to the "correlation is not causality" remark I made in my evaluation of the original manuscript: the authors' answer is reasonable. Still, I would suggest to modify the abstract: "we propose a model of neurodegeneration in which chronic rG4 formation drives proteostasis collapse" => "we propose a model of neurodegeneration in which chronic rG4 formation is linked to proteostasis collapse"All other remarks I made have been answered properly.

We thank the reviewer for their support! We have made the change exactly as requested by the reviewer.